# Qualitative Insights on Barriers to Receiving a Second Dose of Measles-Containing Vaccine (MCV2), Oromia Region of Ethiopia

**DOI:** 10.3390/vaccines12070702

**Published:** 2024-06-22

**Authors:** Kalkidan Solomon, Brooke N. Aksnes, Abyot Bekele Woyessa, Chala Gari Sadi, Almea M. Matanock, Monica P. Shah, Paulos Samuel, Bekana Tolera, Birhanu Kenate, Abebe Bekele, Tesfaye Deti, Getachew Wako, Amsalu Shiferaw, Yohannes Lakew Tefera, Melkamu Ayalew Kokebie, Tatek Bogale Anbessie, Habtamu Teklie Wubie, Aaron Wallace, Ciara E. Sugerman, Mirgissa Kaba

**Affiliations:** 1Department of Preventive Medicine, School of Public Health, College of Health Sciences, Addis Ababa University, Addis Ababa, Ethiopia; mirgissa.kaba@aau.edu.et; 2Global Immunization Division, Centers for Disease Control and Prevention, Atlanta, GA 30329, USA; oqh9@cdc.gov (B.N.A.); xdf2@cdc.gov (A.M.M.); hyy9@cdc.gov (M.P.S.); ccu7@cdc.gov (A.W.); bwf1@cdc.gov (C.E.S.); 3Oromia Regional Health Bureau, Addis Ababa, Ethiopia; abyot.woyessa@tuni.fi (A.B.W.); paulyidu15612@gmail.com (P.S.); tolerabekana@yahoo.com (B.T.); birhanukenate20@gmail.com (B.K.); abebe.bekele@aau.edu.et (A.B.); tesfayegelmesa@yahoo.com (T.D.); 4Ministry of Health of Ethiopia, Addis Ababa, Ethiopia; chagasada@yahoo.com (C.G.S.); yohannes.lakew@moh.gov.et (Y.L.T.); melkamu.ayalew@moh.gov.et (M.A.K.); 5Global Immunization Division, CDC-Ethiopia, Addis Ababa, Ethiopia; 6United Nations International Children’s Emergency Fund, Addis Ababa, Ethiopia; gwako@unicef.org (G.W.); ashiferaw@unicef.org (A.S.); 7African Field Epidemiology Network, Addis Ababa, Ethiopia; btatek@afenet.net; 8Ethiopian Public Health Institute, Addis Ababa, Ethiopia; habtamuteklie2@yahoo.com

**Keywords:** measles, measles vaccine, barrier, qualitative, Ethiopia

## Abstract

**Introduction****:** Ethiopia introduced a second dose of measles-containing vaccine (MCV2) in 2019 to provide further protection against measles and further progress toward elimination. However, the sub-optimal coverage of both MCV1 and MCV2 suggest challenges with vaccine uptake. In this qualitative study, we explored barriers to the uptake of MCV2 among caregivers, community leaders, and healthcare workers (HCWs). **Method:** A qualitative study was conducted between mid-April and mid-May 2021. We selected ten woredas (districts) in the Oromia Region, Ethiopia, stratified by settlement type (urban/rural), MCV1 coverage (high ≥ 80%; low < 80%), and history of measles outbreaks between June 2019 and June 2020. Experiences surrounding barriers to MCV2 uptake were discussed via focus group discussions (FGDs) and in-depth interviews (IDIs) with caregivers of children 12–23 and 24–36 months and key informant interviews (KIIs) with HCWs who administer vaccines and with community leaders. Participants were recruited via snowball sampling. Recorded data were transcribed, translated to English, and analyzed using ATLAS.ti v.09. **Results:** Forty FGDs and 60 IDIs with caregivers, 60 IDIs with HCWs, and 30 KIIs with community leaders were conducted. Barriers among caregivers included lack of knowledge and awareness about MCV2 and the vaccination schedule, competing priorities, long wait times at health facilities, vaccine unavailability, negative interactions with HCWs, and transportation challenges. At the community level, trusted leaders felt they lacked adequate knowledge about MCV2 to address caretakers’ questions and community misconceptions. HCWs felt additional training on MCV2 would prepare them to better respond to caretakers’ concerns. Health system barriers identified included the lack of human, material, and financial resources to deliver vaccines and provide immunization outreach services, which caretakers reported as their preferred way of accessing immunization. **Conclusions:** Barriers to MCV2 uptake occur at multiple levels of immunization service delivery. Strategies to address these barriers include tools to help caretakers track appointments, enhanced community engagement, HCW training to improve provider–client interactions and MCV2 knowledge, and efforts to manage HCW workload.

## 1. Introduction

Every year, nearly 20 million children remain under-vaccinated (missing one or more recommended vaccines) across the globe. Of these, an estimated 13 million have never received a single dose of any vaccine (zero-dose) [1,2]. In 2021, over 85% of these zero-dose children resided in ten countries globally, one of which was Ethiopia, [3,4].

Measles is highly contagious with high morbidity and mortality. However, measles can be prevented through vaccination. Between 2000 and 2018, measles vaccination coverage improved globally and prevented an estimated 23.3 million deaths. Despite the availability of a safe and effective vaccination, measles continues to be a leading cause of mortality and morbidity among young children in Sub-Saharan Africa (SSA). In 2017, WHO recommended that all countries should offer a second routine dose of measles-containing vaccine (MCV2) in the second year of life to provide additional immunity against measles and as a way to improve immunization services beyond the first year of life, including providing catch-up immunization for un-vaccinated or under-vaccinated children [2]. WHO and UNICEF recommend providing MCV2 through routine service initiatives [2].

Ethiopia has been committed to reaching measles elimination since 2001 and has implemented strategies recommended by the World Health Organization (WHO) and UNICEF for accelerating the control of measles, including routine immunization and supplemental immunization activities. To increase MCV2 uptake, the Ethiopian Ministry of Health’s Expanded Programme on Immunization (EPI) introduced MCV2 in February 2019 for children fifteen months of age as part of the routine immunization schedule. In support of MCV2 introduction, measures were taken to strengthen cold chain management, update recording and reporting tools, and train stakeholders [5]. Despite efforts, measles vaccine coverage continues to be sub-optimal with MCV1 coverage at 60% and MCV2 coverage at 46% in 2020. The low population coverage with the recommended 2-dose regimen for measles vaccine has been flagged as a source for concern [6].

Studies show that sociocultural barriers to childhood vaccination in general and the measles vaccine in particular are more pronounced among poor households [7]. However, little is known about the individual perspectives of caregivers and healthcare workers (HCWs) on the barriers for MCV2 uptake in Ethiopia [8,9,10]. Additionally, the Oromia Region is the most populous of Ethiopia’s regional states and city administrations and ranks the third lowest for vaccination coverage, resulting in an estimated >600,000 children under-vaccinated for measles, which also represents approximately half of Ethiopia’s children un-vaccinated for measles [11].

This study therefore aims to explore the barriers which contribute to low MCV2 uptake in selected areas of Oromia to inform future decision making on strategies to strengthen vaccination coverage and equity in Ethiopia.

## 2. Methods

### 2.1. Study Design

The present study employed a qualitative narrative approach to explore barriers to MCV2 uptake in the Oromia Region, Ethiopia. Findings from the quantitative study and detailed site selection methods are reported elsewhere (Woyessa et al.).

### 2.2. Study Area and Period

In brief, 18 of 337 woredas in the Oromia Region were selected for inclusion in the quantitative study after stratifying by setting (urban/rural), MCV1 administrative coverage (high: ≥80%; low: <80%), and recent history of measles outbreak (between June 2019 and June 2020). A further subset of 10 woredas were included in the qualitative assessment reported in this study (Table 1). Data collection for the qualitative assessment was conducted between May and June 2021.

### 2.3. Study Participants

HCWs and caregivers were eligible to participate in the study due to their role in immunization service delivery at the community and facility level and as primary caregivers of one or more children aged 12–23 or 24–36 months old during the study, respectively. Caregivers with the same profile who were not selected for IDIs were recruited for FGDs. Community leaders who were generally believed to have an understanding about children, health, and immunization were involved in KIIs.

Study participants were recruited with support from local health extension workers (HEWs) and through snowball sampling. IDIs and KIIs were conducted in a private space or at participants’ residences, and FGDs were carried out in quiet areas in the community.

Focus group discussions (FGDs), in depth interviews (IDIs), and key informant interviews (KIIs) with either caregivers of children 12–36 months, community leaders, or health care workers (HCWs) were conducted.

IDIs were completed with HCWs who provide immunization services and caregivers of children aged 12–23 and 24–36 months. IDIs with caregivers included questions on the enablers and barriers to the uptake of MCV, while IDIs with HCWs included topics related to the enablers and barriers to MCV service delivery. FGDs were carried out with caregivers of children 12–23 and 24–36 months and included questions about the facilitators and barriers to measles vaccination. KIIs were conducted with local leaders representing both men and women in the community identified as knowledgeable about children, health, and immunization.

### 2.4. Data Collection Guide and Procedures

For all data collection activities, interview guides were developed by the research team to gather detailed perspectives on the barriers for MCV2 uptake at different levels among the interviewed populations. Guides were refined and validated with input from research assistants who were recruited from local study settings during their training for this research. The guides were prepared in English and translated to Amharic and Afan Oromo languages for easy communication based on participants’ preference. The interviews were prearranged, led by the research assistants, and all interviews were conducted face-to-face at a venue of the participants’ choice.

### 2.5. Data Management, and Analysis

Audio recorded data from FGDs, KIIs, and IDIs were transcribed verbatim and translated into English by data collectors. Each transcription and translation was checked for quality and consistency and then imported to ATLAS.ti v.09 for data reduction and analysis. Data collection occurred iteratively with data analysis. The initial coding was made by Kalkidan and Chala, while consistency was checked by Mirgissa and Brooke. Intercoder differences were clarified via discussion, and consensus was reached among the researchers on the codes. The codebook was developed inductively during review of the transcripts. The codes were reduced and placed into logical categories with narrative segments of data to develop themes in line with the objective of the study. In order to intensively describe the findings of this study, the data we used thematic analysis so that the thick description of the findings within the text was the focus. Findings were validated in a consultative workshop with data collectors.

### 2.6. Data Quality Assurance

The interviews were led by trained research assistants who were familiar with the research setting and context. Interviews were conducted in settings preferred by the study participants to create a comfortable environment for discussion. At the end of every interview, member checking was performed to ensure participants’ contributions were correctly captured. Daily briefs were also held with the team members to ensure the quality of data and address any challenges as they emerged. Data were read and re-read and coded by two of the researchers (Kalkidan and Chala), while two other research team members checked the consistency of codes (Mirgissa and Brooke). Data were triangulated by method, place, and participant profile.

### 2.7. Ethical Considerations

All participants were ≥18 years of age and provided verbal informed consent to participate. The protocol was approved by the scientific and ethical review board of the Oromia Regional Health Bureau, Addis Ababa, Ethiopia, and the U.S. Centers for Disease Control and Prevention (CDC) deemed that this activity was an assessment of public health practice and as such determined it as non-research.

## 3. Findings

### 3.1. Sociodemographic Profile of Participants

In total, sixty HCWs and sixty caregivers participated in IDIs. Forty FGDs were organized with two groups (caregivers with one child aged 12–23 or 24–36 months and caregivers with more than one child in this age range), and 30 KIIs were conducted with community leaders. The characteristics of the participants are summarized in Table 2.

The study’s findings are presented in three different themes and sub-themes, namely: (1) Caregiver-level barriers, which included low MCV2 awareness, competing priorities, transportation difficulties, and the need for repeated trips to the health facility; (2) Community-level barriers, which included misconception toward MCV2, competing priorities, low awareness about MCV2, and perception to MCV2; and (3) Health system-level barriers, which encompassed workload, insufficient capacity building, demotivation, and the rule to open the vial (Table 3). To ensure the quality of this study, the researchers used specific phrases while writing the findings of the interview.

### 3.2. Caregiver-Level Barriers

Low awareness of MCV2 among caregivers was observed in all participating woredas and across all sociodemographic characteristics. A prominent sentiment was that MCV2 was a new vaccine for which caregivers did not know or remember the timing or purpose.

“*We never received information about this new vaccine (MCV2). I am just hearing this from you*”(Woman with a child 12–23 months, FGD, Dhas).

“*I know the values of vaccinating children (…). However, I do not know about this new vaccine [MCV2]. I do not understand its purpose and why it is given after children complete the known course of vaccination*”(Women with a child 24–36 months, IDI, Haromaya).

“*I was told by the provider about the next vaccination schedule [MCV2] when I took my child to get the last vaccine at age nine months. I forgot this all together. I realized most caregivers forget this round of vaccine. We know and remember the nine-month vaccination while this coming only after one year and five months is easy to forget. Nobody reminded me of this*”(Woman with a child 12–23 months, IDI, Mendi).

“*I do not know why the schedule of MCV2 is different from what I know about other known vaccines. Health workers appointed me to bring my child for vaccination at nine months but not at fifteen months*”(Woman with a child 12–23 months, IDI, Shashemene).

Caregivers also reported that competing priorities such as household chores and expectations to participate in social events were barriers to seeking MCV2 vaccination.

“*We live in rural area where we have a lot of responsibilities. We look after dairy cattle, search for water, collect firewood, look after the children and elderly but are also expected work in the farms. There are also social expectations that affect taking children to health facilities—even in other times*”(Woman with a child 24–36 months, FGD, Midakegn).

For caregivers, transportation and distance to their dedicated health centers were also discussed as barriers by respondents.

“*Mothers who live in the remote rural kebele often do not take their children to health centers due to irregular transportation services. They prefer to go to a closer woreda, but they are denied service there as they are not within the catchment area*”(HEW, IDI, Bishan Guracha).

Caregivers also reported a lack of motivation to go or return to health facilities for MCV2 due to the unavailability of immunization services, long wait times, and unpleasant interactions with HCWs.

“*I was advised on the value of MCV2 vaccine and encouraged by the HEW in our village to take my child to the health center. At the health center, (…) I was told to come back another time. I did not come back next time since I was busy and was also embarrassed by the treatment from the health worker*”(Woman with a child 24–36 months, KII, Shashamane).

“*It is unfortunate that I came to the health center for this new vaccine. After waiting for almost for half a day, I was told there is no vaccine. They told me to go to other health center, where I could find the vaccine, which is far away, and I couldn’t do that*”(Woman with a child 12–23 months, IDI, Haromaya).

### 3.3. Community-Level Barriers

Although community opinion leaders and caregivers across the study settings argued that immunization services are well recognized and accepted as valuable for children, lack of clear information about MCV2 and its benefits has resulted in misconceptions surrounding the vaccine.

“*We don’t know why this new one [MCV2] was brought. We know MCV2 is a new vaccine, and there are rumors around that it may affect the health the child*”(Community leader, KII, Bishan Guracha).

“*This new vaccine is not given equal attention by health professionals in providing information about the vaccine to the public. There seems to be something secret about MCV2 which is the source of suspicion about the vaccine*”(Woman with a child 12–23 months, IDI, Burayu).

MCV2 was considered as an add-on to the existing schedule. Providers reported that, at the time of the survey, immunization registers did not have a space to enter data about MCV2 vaccination, reinforcing the need to improve the integration of MCV2 as part of the routine immunization program, including reporting tools.

“*…even the current registration lacks section for recording information concerning MCV2*”(Healthcare worker, IDI, Mendi).

“*I am unaware of this new vaccine (MCV2), though. I don’t know why it’s given to kids after they have received the recognized series of vaccinations, and I don’t know why it added*”(Woman with a child 24–36 months, IDI, Midakegn).

“*This new vaccine is not considered as mandatory in the community, so we don’t want to put additional pressure on ourselves*”(Woman with a child 24–36 months, FGD, Midakegn).

### 3.4. Barriers at the Healthcare System-Level

Analysis also revealed a number of health facility-related barriers including insufficient numbers of trained service delivery staff and heavy workloads.

“*No one understands the pressure under which we operate. We have a shortage of staff at antenatal care, family planning, and especially we have a problem with the EPI. Since we operate with minimum number of staff, we cannot vaccinate children as they report to the health facility even if we have the supplies*”(Healthcare worker, IDI, Bishan Guracha).

It was reported that HEWs in the community were overwhelmed by the size of their catchment areas, making it difficult to carry out planned MCV2 provision and related activities which were viewed as new or extra tasks.

“*We are responsible to routinely visit households to discuss with women on various health issues. This is in addition to the provision of services at the health post. It is cumbersome for an extension worker to do every new initiative which we don’t even know about*”(HEW, IDI, Bishan Guracha).

“*Covering a kebele with multiple villages and such a large number of households by two health extension workers is difficult with existing activities let alone with an additional task*”(Healthcare worker, IDI, Lode Hetosa).

Participants reported a lack of a shared understanding about MCV2 services among HCWs which HCWs attributed to a lack of in-service training and available job aids on MCV2.

“*For such services as family planning, HIV/AIDS and even regarding polio vaccination, we got training and have job aids. Meanwhile for MCV2, which has started recently, care providers both at community level and those of us at the health facility did not get any training on MCV2, and there is no job aid to guide service provision*”(Healthcare worker, IDI, Dodola).

It was also reported that providers have a difficult time deciding to open a vaccine vial if a minimum number of children, usually five, were not present. One of the participants noted that,

“*Even if MCV2 vaccine is available, there is no guidance on for how many children to open the vial. As a result, caregivers may be appointed to come back another time, and they do not come back. Personally, I do not mind providing vaccine to every child as they come. However, what could happen if there are no other children coming for vaccination and if the vial is wasted? This is a major concern for all the providers and there is no guidance*”(Healthcare worker, IDI, Lode Hetosa).

## 4. Discussion

The findings from this study depict multiple barriers operating in tandem to MCV2 vaccine uptake in the Oromia Region of Ethiopia two years post-MCV2 introduction at individual, community, and healthcare system levels.

The widespread lack of caregiver awareness of MCV2 was found to be a major barrier to uptake identified by all participant types, which is consistent with studies in other regions of Ethiopia [6,12]. It is possible that the lack of information and awareness of the vaccine caused or compounded the reported misconceptions and rumors in the community. This is supported by studies conducted globally in low-, middle-, and high-income countries that have attributed limited awareness and consequent misconceptions both at individual and community levels to low MCV2 coverage [13,14,15]. The lack of caregiver awareness of MCV2 identified may be compounded by the reported limitations of MCV2 knowledge and skills to capacitate caregivers among trusted sources of information (HCWs and community leaders). Since the cascaded training of HCWs was carried out and training manuals were developed during the initial rollout of MCV2, strategies for optimization should be explored in the future, and how high HCW turnover and translation to the HEW level can be taken into consideration.

Another prominent barrier to MCV2 uptake found at caregiver, community, and healthcare system levels is that the second dose of the measles vaccine is perceived as relatively unimportant. A lack of information and education for all stakeholders, stockouts, no official recording and reporting forms, and a widespread perception that immunization is complete by 12 months of age among the community undermine the idea that MCV2 is a routine vaccine that is important for the health of children. According to the health belief model, if the perceived benefit of adopting a health intervention is low, people are less likely to adopt it [16]. This may somewhat explain the results seen here. Similar findings have been published for MCV2 and other vaccines provided in the second year of life such as the meningococcal conjugate vaccine in other African countries [17].

As the above lack of the perceived importance of MCV2 seems to originate from across the immunization ecosystem, steps to build a perception of the importance of the MCV2 vaccine need to occur at all levels. HCWs and community leaders might receive enhanced training and engagement on MCV2 and its purpose, an intervention that has been observed in existing research to correlate with increased immunization uptake [8,13,15]. Additionally, providing healthcare workers and community leaders training on communication/counseling skills and targeted literacy-level-appropriate messages about MCV2 may in turn increase community awareness of MCV2. Since messaging was created nationally at the time of introduction, assessment of the quality, reach, and implementation of such messaging needs to be more real time in new vaccine introductions in the second year of life. Updating official recording and reporting forms to include MCV2 may also reinforce its importance as a routine immunization [2] and should be carried out well in advance of introduction to avoid confusion and to provide time for the orientation of HCWs.

It is widely documented that caretakers’ negative experiences at health facilities can hinder them from seeking health services, including for MCV2 vaccination for their child [18]. Here, caretakers reported that impolite interactions with HCWs, long wait times, and the cumbersome need to make multiple trips to the health facility due to the unavailability of the vaccine made them less likely to seek timely MCV2. Successful interventions targeting improved experiences at healthcare facilities during vaccination visits in the second year of life have included improving waiting areas, training healthcare workers on client communications, and implementing appointment systems [18,19]. Future research to determine the effectiveness of these interventions at scale will be useful.

Further, caretakers’ reported multiple trips to a health facility may be exasperated by HCWs’ hesitancy to open a multidose vaccine vial without a certain number of children present for fear of wastage or reprimand, since a ten-dose measles vaccine vial was in use at the time of the study [20]. Although contrary to Ethiopian and WHO vaccine management guidelines which state multidose vials should be opened even if only one child is present, similar concerns among HCWs have been documented in Ethiopia and other low-income settings [21]. Providing HCWs with clarity on when to open a multidose measles vaccine vial is essential, alongside alleviating their fears around high vaccine-wastage repercussions.

Respondents noted that limited motivation, competence, and the heavy workload of HCWs hindered the uptake of MCV2. Studies in other countries have also captured this association and its link to financial constraints [5,8,13,18]. When HCWs are not provided adequate support structures, refresher training, and routine supportive supervision, health services may be impacted. Since high HCW turnover and re-assignment of HCWs to different health facilities/woredas occurred in the region during the years around MCV2 introduction, the impact of this generalized infrastructure change needs to be taken into consideration for new vaccines, especially those in the second year of life.

These findings present significant implications for vaccine equity. Research shows that less educated caretakers are less likely to utilize immunization services, including MCV2 [22,23], and this study shows a widespread lack of awareness of MCV2 among the population. When equipping HCWs and HEWs to increase awareness, attention must be given to ensure communication with caretakers happens at the appropriate level of literacy and linguistic competency so that these groups are equitably sensitized. Additionally, immunization outreach services are most needed by those with the least available time and resources to travel to a health facility. Some woredas cited difficulties in providing outreach services due to human resource gaps, thus creating pockets of missed communities. Solutions should be explored, including bringing in health personnel from outside woredas for outreach activities, to ensure the hardest-to-reach children have the opportunity to receive MCV2. Finally, the opportunity costs of seeking immunization create a disproportionate burden for communities at already high risk of missing immunizations. The findings here identified multiple factors increasing the opportunity costs of MCV2 immunization, including long wait times at health facilities and needing to return multiple times to the health facility, consistent with findings across Sub-Saharan Africa [24]. Efforts to create a more efficient and quality immunization experience for caretakers can help reduce these opportunity costs.

This study is subject to a number of limitations. Child health is a joint concern for the mother and father, and one major limitation is that no male caretakers were included in these discissions. This may potentially limit the range of views captured here. The fact that macro-level health managers were not involved may have also resulted in limited evidence on logistics and procedures.

## 5. Conclusions

This study underscores that MCV2 uptake is affected by barriers that work in tandem at multiple levels. Individual-level barriers included misconceptions in connection to MCV2 as a new addition to established routine immunizations; a limited effort to capacitate HCWs to better respond to caretakers’ needs and a lack of available job aids and guidance coupled with limited logistic preparedness affected MCV2 service uptake. Strengthened MCV2 uptake calls for a multi-pronged and re-invigorated strategic focus at the individual, community, and primary healthcare level.

## Figures and Tables

**Table 1 vaccines-12-00702-t001:** Woredas selected for qualitative study.

S. No	Zone	Performance	Profile	Performance
1	Arsi	Lode Hetosa	Rural	Low MCV1 Coverage
2	Bishan Guracha Town	Bishan Guracha Town	Urban	Low MCV1 Coverage
3	Borena	Das	Rural	High MCV1 Coverage
4	Burayu Town	Burayu Town	Urban	High MCV1 Coverage
5	East Hararge	Haromaya Town	Urban	Low MCV1 Coverage
6	East Shewa	Fentale	Rural	Low MCV1 Coverage
7	West Arsi	Dodola Rural	Rural	Measles Outbreak
8	West Wellega	Mendi Town	Urban	High MCV1 Coverage
9	Shashemene Town	Shashemene Town	Urban	Measles Outbreak
10	West Shewa	Mida Kegni	Rural	High MCV1 Coverage

**Table 2 vaccines-12-00702-t002:** Characteristics of the study participants, Oromia Region, Ethiopia.

Characteristics of Participants	Method of Data Collection			
	FGD (Caregivers) N = 240 Participants (40 FGDs)	KII (Community Leaders) N = 30	IDI (Caregivers) N = 60	IDI (HCWs) N = 60
**Sex**				
**Male**	-	16	-	8
**Female**	40 (sessions)	14	60	32
**Age of participant**				
**19–24**	24		13	22
**25–30**	96	6	32	26
**31–36**	90	6	11	12
**37–42**	18	12	4	
**43–48**	12	2		
**48+**		4		
**Occupation**				
**Housewife**	25		45	
**Farmer**	8		10	
**Teacher**	3			
**Merchant**	4		5	
**HCWs**				60
**Community leader**		30		
**Educational status**				
**No formal education**	19		52	
**Elementary education**	17	14	6	
**High school**	2	5	2	
**Diploma**	2	7		44
**First degree**		4		16

**Table 3 vaccines-12-00702-t003:** Barriers to receiving a second dose of measles-containing vaccine (MCV2).

Level of Barrier	Barrier Identified
*Caregiver-level barriers*	*Low MCV2 awareness*
	*Competing priorities*
	*Transportation difficulties*
	*Need for repeated trips to health facility*
	*Poor communication by HCWs*
*Community-level barriers*	*Misconception*
	*Competing priorities*
	*Low awareness*
	*Perception to MCV2*
*Health system-level barriers*	*Workload*
	*Insuffient capacity building*
	*Demotivation*
	*Rule to open the vial*

## Data Availability

All relevant data are within the manuscript, while the raw data will be available upon reasonable request from the corresponding author.

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
