# Peer review of "Qualitative Insights on Barriers to Receiving a Second Dose of Measles-Containing Vaccine (MCV2), Oromia Region of Ethiopia"

_vaccines, 2024, doi:10.3390/vaccines12070702_

Round 1

Reviewer 1 Report

Comments and Suggestions for Authors

This important study aims to conduct a qualitative study for exploring barriers to the uptake of MCV2 among caregivers, community leaders, and healthcare workers (HCW) in Oromia Region of Ethiopia. The manuscript, however, lacks some important information in the main text. Here are some points I would like the authors to consider to further highlight the contribution of the study.

1.     Introduction. The introduction section of the article is too long, please shorten it and just emphasize the main research question and purpose.

2.     Introduction. Please explain the reason why do you choose MCV as the study object, not other childhood vaccines, such as DTP or HepB?

3.     In the Data collection technique and procedures section, how do you choose the participants and how to calculate the sample size? Please add more details.

4.     In the method section, how do you design the qualitative questionnaire and how to evaluate the quality of your question lists? Please provide more details or provide the question lists in the appendix.

5.     In Table 2, this table should be three-line table.

6.     In the result section, could you please summarize your main results into a table or figure. The long paragraph is hard to read.

7.     In the discussion section, you should compare your results with other studies, especially with other low-and-middle-income country study.

Author Response

Reviewer: Comments and Suggestions for Author

This important study aims to conduct a qualitative study for exploring barriers to the uptake of MCV2 among caregivers, community leaders, and healthcare workers (HCW) in Oromia Region of Ethiopia. The manuscript, however, lacks some important information in the main text. Here are some points I would like the authors to consider to further highlight the contribution of the study.

  1. The introduction section of the article is too long, please shorten it and just emphasize the main research question and purpose.

  • Thank you, we tried to shorten the introduction section as in the revised the manuscript.

  1. Please explain the reason why do you choose MCV as the study object, not other childhood vaccines, such as DTP or HepB?

  • Thank you, we included the justification on why we chose MCV2 (Introduction section). We explain that despite the availability of a safe and effective vaccination, measles continues to be a leading cause of mortality and morbidity among young children in Sub-Saharan Africa. Plus, due to ongoing and significant outbreaks of measles in Ethiopia, and the associated morbidity and mortality, it is of critical importance to understand the barriers to the low uptake of the vaccine in this context to inform programmatic changes.

  1. In the Data collection technique and procedures section, how do you choose the participants and how to calculate the sample size? Please add more details.

  • Thank you, we tried to revise the data collection technique and procedure section based on the suggestion. However, to note is that in qualitative research, we choose study participants based on a predefined profile that make them well positioned to address the research question instead of the statistical type of sampling used in quantitative research offering equal chance for study population to participate (Method section). We feel this approach adds value and is in line with standard qualitative research methodology.

  1. In the method section, how do you design the qualitative questionnaire and how to evaluate the quality of your question lists? Please provide more details or provide the question lists in the appendix.

  • Thank you, we revised the quality assurance section to provide further detail on evaluation of data quality. Please also see lines 151-156 in the methods section which describes how the data collection tools were designed and evaluated.

  1. In Table 2, this table should be three-line table.

  • Thank you we revised it (Table 2)

  1. In the result section, could you please summarize your main results into a table or figure. The long paragraph is hard to read.

  • Thank you. We have added a table to summarize the main finding headings (Table 3), but would note that qualitative data is narrative in nature and does not receive any quantification during analysis. To be consistent with generally recommended processes for representation of qualitative findings, the narrative is a necessity for interpretability.

  1. In the discussion section, you should compare your results with other studies, especially with other low-and-middle-income country study.
  • Thank you we cite more recent studies conducted in Africa (Discussion and reference section)

Reviewer 2 Report

Comments and Suggestions for Authors

There is a great deal to commend in this study.  It looks for barriers to Measles innoculation at multiple levels of concern-the individual, the community level and the institutional level (health care workers).  The sampling strategy is well designed and commendable.  The results also have considerable pragmatic value-enhanced community outreach activities should be prioritized as a result of these findings. 

Here are several things that deserve some modification and enhancement in the report.  In the introduction, the research team should provide a clear explanation of why the MCV2 is clearly superior to the MCV1 alone.  This can be done in a couple of sentences.   My bigger concern is that the results can be communicated more strongly.  Specifically, I would like to see information on the percentages of IDI's and Focus groups that mentioned each of the major themes.  This will give us some idea of the relative salience/importance of each theme at the individual level and at the community level.  The unit of analysis here, is mention in the individual unit interview of either the individual interviews or the focus group discussions.  This information could be used to identify the rank order of importance of concerns at each level.    This information can be incorporated in charts and will further serve the purposes of data reduction and comprehension for a very extensive and complex data set.  Solid work overall.

Author Response

Reviewer: Comments and Suggestions for Author

There is a great deal to commend in this study.  It looks for barriers to Measles inoculation at multiple levels of concern-the individual, the community level and the institutional level (health care workers).  The sampling strategy is well designed and commendable.  The results also have considerable pragmatic value-enhanced community outreach activities should be prioritized as a result of these findings. 

  • Thank you for your comments and the encouragement.

Here are several things that deserve some modification and enhancement in the report.  In the introduction, the research team should provide a clear explanation of why the MCV2 is clearly superior to the MCV1 alone.  This can be done in a couple of sentences.  

  • Thank you, we further explained the relative importance of MCV2 in light of the purpose of the study (introduction section). Namely, MCV2 provides a higher immunity against measles for children, and offering a vaccine in the second year of life for a child provides the health system with an additional contact point to identify any missing vaccine doses and provide other essential routine health services.

My bigger concern is that the results can be communicated more strongly.  Specifically, I would like to see information on the percentages of IDI's and Focus groups that mentioned each of the major themes.  This will give us some idea of the relative salience/importance of each theme at the individual level and at the community level. The unit of analysis here, is mention in the individual unit interview of either the individual interviews or the focus group discussions.  This information could be used to identify the rank order of importance of concerns at each level.    This information can be incorporated in charts and will further serve the purposes of data reduction and comprehension for a very extensive and complex data set.  Solid work overall.

  • Thank you for your comments and suggestion. We have added, and summarizing the themes identified during analysis to help make the results easier to digest (table three). The unit of analysis for this study is at multiple level for barriers were analyzed at individual, institutional and community level. So, findings for each level have come from IDIs that were run at individual care giver’s level (60), and HCWs (60), FGDs (40 sessions) with care givers and KII (30) with community leaders which is illustrated in the manuscript. Plus, it may be with noting that this paper is a tandem paper in the same issue of this supplement where we side by side will aim to publish the quantitative study findings from this work. We strongly feel that this is very worthwhile as the qualitative work reported was extensive and stands along while fully complementing the quantitative findings (in the other paper entitled Factors associated with uptake of routine second dose of measles-containing vaccine among young children, Oromia Regional State, Ethiopia, 2021” by Woyessa et al.).

Reviewer 3 Report

Comments and Suggestions for Authors

Thank you for the opportunity to review your article. Please see my comments attached- 

Peer review- submitted to Vaccines

Qualitative insights on barriers to receiving a second dose of 2 measles containing vaccine (MCV2), Oromia Region of Ethiopia

Originality / Novelty

The paper is very well written, very logically constructed and the aims are clearly outlined and supported by the background. The authors are “splitting” out the qualitative aspects of their mixed methods project in Ethiopia. This is an important study specific to their country as it evaluated subject vaccine hesitancy issues. Although focused on issues within Ethiopia there may be translatable elements to our regions and countries.  One area that may be improved is a more robust description of the government procedures in place to educate and promote vaccination and after the initiation to improve vaccine uptake. That is what are the usual public health administrative procedures and processes utilized by Ethiopia as a potential area to improve?

Significance of Content

Highly significant for the country and relevant as it focuses on specific actionable findings from the well conducted interviews. As stated, these impediments could be translated to other countries with similar vaccine challenges. However, what was the role of the Ethiopian health authority efforts prior to the study to educate and promote the measles vaccination. This is an area that would need to be evaluated for possible improvement in their approach

Quality of Presentation

The quality of the presentation is outstanding. As stated well written very logical constructed manuscript and easy to follow for general readership. All relevant concerns were discussed in appropriate detail including sampling, human research subject protections, development and validation of the interview and interview teams for the qualitative presentations and data collection

Scientific Soundness

The investigators followed standard and accepted procedures for qualitative research as described. Presentation of findings was insightful and appropriate

 Interest to the readers

In general, interesting to discover the impediments for vaccine understanding and acceptance in this area of Africa. The qualitative findings I believe would have interest to public health officials outside of Ethiopia and translate to what solutions could be instituted.

Overall Merit

High merit within the region and Africa but potentially other similar areas. The project does inform all of us in public health of the possible similar challenges.

Are the references cited in this manuscript appropriate and relevant to this research?

 Yes

Author Response

Reviewer: Comments and Suggestions for Author

Thank you for the opportunity to review your article. Please see my comments attached- 

Peer review- submitted to Vaccines

Qualitative insights on barriers to receiving a second dose of measles containing vaccine (MCV2), Oromia Region of Ethiopia

Originality / Novelty

The paper is very well written, very logically constructed and the aims are clearly outlined and supported by the background. The authors are “splitting” out the qualitative aspects of their mixed methods project in Ethiopia. This is an important study specific to their country as it evaluated subject vaccine hesitancy issues. Although focused on issues within Ethiopia there may be translatable elements to our regions and countries.  One area that may be improved is a more robust description of the government procedures in place to educate and promote vaccination and after the initiation to improve vaccine uptake. That is what are the usual public health administrative procedures and processes utilized by Ethiopia as a potential area to improve?

  • Thank you, we include the public health strategies considered by Ethiopian Ministry of Health to improve vaccine up take (Introduction section)

Significance of Content

Highly significant for the country and relevant as it focuses on specific actionable findings from the well conducted interviews. As stated, these impediments could be translated to other countries with similar vaccine challenges. However, what was the role of the Ethiopian health authority efforts prior to the study to educate and promote the measles vaccination. This is an area that would need to be evaluated for possible improvement in their approach

  • Thank you, as suggested we explained the role of Ethiopian health authority effort (Introduction section)

Quality of Presentation

The quality of the presentation is outstanding. As stated well written very logical constructed manuscript and easy to follow for general readership. All relevant concerns were discussed in appropriate detail including sampling, human research subject protections, development and validation of the interview and interview teams for the qualitative presentations and data collection

  • Thank you for the encouragement

Scientific Soundness

The investigators followed standard and accepted procedures for qualitative research as described. Presentation of findings was insightful and appropriate

  • Thank you

 Interest to the readers

In general, interesting to discover the impediments for vaccine understanding and acceptance in this area of Africa. The qualitative findings I believe would have interest to public health officials outside of Ethiopia and translate to what solutions could be instituted.

  • Thank you  

Overall Merit

High merit within the region and Africa but potentially other similar areas. The project does inform all of us in public health of the possible similar challenges.

  • Thank you

Are the references cited in this manuscript appropriate and relevant to this research?

 Yes

  • Thank you

Round 2

Reviewer 1 Report

Comments and Suggestions for Authors

None

Reviewer 2 Report

Comments and Suggestions for Authors

I am satisfied with your responses to my review.